# Improvement in the Sustained-Release Performance of Electrospun Zein Nanofibers via Crosslinking Using Glutaraldehyde Vapors

**DOI:** 10.3390/foods13101583

**Published:** 2024-05-20

**Authors:** Shumin Wang, Jingyu Li, Pengjie Wang, Ming Zhang, Siyuan Liu, Ran Wang, Yixuan Li, Fazheng Ren, Bing Fang

**Affiliations:** 1Key Laboratory of Precision Nutrition and Food Quality, Department of Nutrition and Health, China Agricultural University, Beijing 100083, China; sdutshumin@163.com (S.W.); wpj1019@cau.edu.cn (P.W.); siyuan.liu@cau.edu.cn (S.L.); wangran@cau.edu.cn (R.W.); liyixuan@cau.edu.cn (Y.L.); renfazheng@263.net (F.R.); 2School of Food and Health, Beijing Technology and Business University, Beijing 100048, China; ljy110808@163.com (J.L.); zhangming@th.btbu.edu.cn (M.Z.); 3Food Laboratory of Zhongyuan, Luohe 462300, China

**Keywords:** electrospinning, zein, nanofibers, crosslinking, sustained release

## Abstract

Volatile active ingredients in biopolymer nanofibers are prone to burst and uncontrolled release. In this study, we used electrospinning and crosslinking to design a new sustained-release active packaging containing zein and eugenol (EU). Vapor-phase glutaraldehyde (GTA) was used as the crosslinker. Characterization of the crosslinked zein nanofibers was conducted via scanning electron microscopy (SEM), mechanical properties, water resistance, and Fourier transform infrared (FT-IR) spectroscopy. It was observed that crosslinked zein nanofibers did not lose their fiber shape, but the diameter of the fibers increased. By increasing the crosslink time, the mechanical properties and water resistance of the crosslinked zein nanofibers were greatly improved. The FT-IR results demonstrated the formation of chemical bonds between free amino groups in zein molecules and aldehyde groups in GTA molecules. EU was added to the zein nanofibers, and the corresponding release behavior in PBS was investigated using the dialysis membrane method. With an increase in crosslink time, the release rate of EU from crosslinked zein nanofibers decreased. This study demonstrates the potential of crosslinking by GTA vapors on the controlled release of the zein encapsulation structure containing EU. Such sustainable-release nanofibers have promising potential for the design of fortified foods or as active and smart food packaging.

## 1. Introduction

Microbial contamination is one of the main causes of food spoilage and poisoning during the food storage period [1]. Currently, the application of chemical fungicides is the most commonly used preservation approach. However, these chemicals have potential hazards to human health and the natural environment [2]. Therefore, it is essential to develop safe, eco-friendly, and effective natural agents for food preservation. Eugenol (EU) is the key component of clove oil and possesses remarkable pharmacological actions, such as inflammatory, antiflatulent, anticonvulsant, antipyretic, antitumor, pain-relieving, and neuroprotective effects [3,4,5]. Moreover, EU has shown strong activity against pathogenic viruses, bacteria, and fungi, including *Staphylococcus aureus*, *Escherichia coli* O157:H7, *Salmonella enterica*, *Helicobacter pylori*, *Campylobacter jejuni*, and *Listeria monocytogenes* [6,7,8,9], and hence represents a promising alternative to chemical fungicides. However, the intense aroma, high volatility, and easy decomposition through light, oxidation, and heat processes of EU greatly affect its stability, making it less effective for food preservation [2,10,11]. Therefore, designing appropriate encapsulation systems can overcome these limitations and protect EU for better physicochemical stability and functional activity.

Electrospinning is a cheap and well-established technology that does not involve severe conditions (extreme temperature or pressure); therefore, it has great potential for encapsulating and delivering sensitive bioactive compounds [2]. In addition, the obtained nanofibers possess a high load efficiency, high porosity, and specific surface area, as well as a controllable morphology, which can stabilize and adjust the release behavior of active compounds [12]. Numerous synthetic and biopolymers can be electrospun to produce nanofibers. Compared with synthetic polymers, natural biopolymers have received much interest because of their nontoxicity, renewability, and biocompatibility [8,12]. Nevertheless, unlike stable nanofibers made of synthetic polymers, volatile bioactives loaded in electrospun biopolymer nanofibers are prone to burst and uncontrolled release in water, which results in poor bioaccessibility of the active additives. Therefore, the development of biopolymer nanofibers with a controlled release nature is crucial to enhance food sustainability and decrease waste.

Coaxial electrospinning [13,14], polymer blending [15,16], and crosslinking [17] have been used to enhance the release behavior of functional components in biopolymer nanofibers. In the literature, coaxial electrospinning has been successfully used to produce core–shell biopolymer nanofibers with a better-controlled release performance [13,14,18,19]. However, electrospinning’s shortcomings lie in the slow production rate, high process requirements, and poor compatibility of core–shell fluid [20,21]. Moreover, it was found that the release profiles of functional components from biopolymer nanofibers could be tuned by blending with polymers [15,22]. Most synthetic polymers are generally non-biodegradable or slow-degradable and need to be dissolved in toxic organic solvents, which restrict the applications of nanofibers in food fields [22,23]. Good compatibility and the homogeneous partitioning of synthetic polymers with biopolymers are also challenging [24]. In addition, crosslinking is an interesting method that can be used to reinforce the structure of biopolymer nanofibers, thus improving their release properties [25,26,27,28]. In comparison, glutaraldehyde (GTA) is an effective crosslinking agent, which has a low cost, fast reaction time, and the ability to react with the amine or hydroxyl groups of protein molecules [29]. However, GTA is toxic and may remain in electrospun nanofibers [26,30]. To overcome this limitation, vapor-phase crosslinking and vacuum drying post-treatment can be performed, which have been shown to have little or no cytotoxic effects [31,32,33]. Several studies have investigated the effect of GTA vapor crosslinking on the release performance of electrospun nanofibers [34]. Baykara et al. (2022) reported that the vapor-phase crosslinking of GTA on PVA/gelatin nanofibers did not significantly affect the release of gentamicin [35]. Hadyz et al. (2021) reported that chitosan/PEO nanofibers crosslinked by vapors of GTA extended the release time of nizatidine [33]. However, to the best of our knowledge, current reports mainly focus on synthetic polymers. Studies on the effect of crosslinking by vapor GTA on the release profile, stability, and structural properties of biopolymer nanofibers are still limited.

Zein, a byproduct of the bioethanol industry, is the main storage protein in maize and is recognized as a GRAS polymer [36]. Zein has good fiber formation properties and can fabricate nanofibers using nontoxic organic solvents, making it an ideal biopolymer for electrospinning and nanofiber formation [37,38]. Zein is water-insoluble due to the rich nonpolar amino acids in its structure, which makes it suitable for effectively entrapping hydrophobic functional ingredients into nanofibers, contributing to good compatibility and potential interactions [39]. The encapsulation of many bioactive compounds in zein matrices has been reported, demonstrating their increased stability and functional properties [36,40,41,42]. Furthermore, zein has shown low digestibility and controlled release capability [43,44]; thus, its nanofibers have drawn great interest in the controlled delivery of bioactive compounds. Silva et al. (2020) indicated a lower release rate of tryptophan from zein nanofibers compared with that from zein films [45]. In the study by Defrates et al. (2021), a more sustained release of model drugs (rifampin, rhodamine B, and crystal violet) was observed in zein fibers compared with zein films [46]. Nevertheless, limited studies have been conducted on the sustained or controlled release of volatile functional compounds in zein nanofibers.

Developing biodegradable active packaging with a sustained-release nature for volatile bioactives is of great interest in food preservation and transportation. Therefore, the purpose of the present work was to overcome the limitations of fast and uncontrolled release of volatile actives in biopolymer nanofiber materials for developing slow-release zein nanofibers via crosslinking using GTA in the vapor phase. The influence of crosslinking on the physicochemical properties, including the fiber morphology, water resistance, and mechanical properties, of zein nanofibers was investigated. To analyze the crosslinking mechanism between GTA and zein, the intermolecular interactions were studied using spectroscopic methods. EU was added to zein nanofibers and then crosslinked by GTA vapors. The effect of crosslinking times on the EU release behavior and corresponding release mechanism was assessed. The encapsulation and in vitro release tests of EU suggest great potential for sustained-release zein nanofibers in the food and pharmaceutical industries.

## 2. Materials and Methods

### 2.1. Materials

Zein was purchased from Sigma-Aldrich (St. Louis, MO, USA). Glacial acetic acid, eugenol (EU), and glutaraldehyde (GTA) were purchased from Sinopharm Chemical Reagent Co., Ltd. (Beijing, China).

### 2.2. Fabrication and Crosslinking of Electrospun Zein Nanofibers

The spinning solution was prepared by dissolving zein (30%, *w*/*v*) in acetic acid at 25 ± 2 °C under magnetic stirring. After standing for 20 min to remove bubbles, the solution was then electrospun using an electrospinning device (HZ-11, Huizhi Electrospinning (HZE), Qingdao, China) equipped with an applied voltage of 16 kV at a constant flow rate of 1 mL/h, a target roll speed of 1000 rpm, and a needle tip to collector distance of 10 cm. The spinning solution volume of each sample remained constant at 2 mL, and the nanofibers were collected on aluminum foil. The electrospinning zein nanofibers were labeled GTA_0h.

Electrospun zein nanofibers were crosslinked with GTA vapors. Specifically, an aqueous GTA solution (25%, *v*/*v*) was placed into an airtight desiccator, with the zein nanofibers placed in the headspace. The entire desiccator was maintained at 20 ± 2 °C for 3, 6, 9, and 12 h. After crosslinking, the nanofibers were placed in a vacuum drying oven at 40 °C for 12 h to remove the remaining GTA. The crosslinked zein nanofibers for different times were referred to as GTA_3h, GTA_6h, GTA_9h, and GTA_12h.

### 2.3. Scanning Electron Microscopy (SEM)

The zein nanofiber samples were coated with a gold layer under vacuum, and their morphology was observed using a scanning electron microscope (Hitachi, Tokyo, Japan). The fiber diameters were measured using ImageJ (x64) software based on 200 random fibers of each sample [31].

### 2.4. Mechanical Properties

The mechanical properties of the zein nanofiber samples were determined on a universal testing machine (Instron, Norwood, MA, USA) according to the method of Lu et al. (2017) with some modifications [47]. Each sample was cut into 1 cm × 3 mm (length × width) pieces. Before starting the measurement, the thickness of each nanofiber sample was examined using micrometer calipers (KAFUWELL YB5001A, Hangzhou, China). A crosshead speed of 5 mm/min and a 5 mm gauge length were used during the measurement. The test was repeated five times for each sample. The tensile strength (TS), elongation at break (EB), and Young’s modulus (YM) were calculated using the following formulas:TS = *F_m_*/*S*(1)
EB = (*L_b_* − *L*_0_)/*L*_0_ × 100%(2)
YM = (*F_m_* × *L_g_*)/(*S* × *L_m_*)(3)
where
*F_m_* = maximum load (N) recorded*S* = cross-sectional area of the nanofibers*L_b_* = length (mm) at the breaking point*L*_0_ = initial length (mm) of the nanofibers*L_m_* = test length (mm) corresponding to the maximum load*L_g_* = gauge length (mm)


### 2.5. Water Contact Angle (WCA)

The WCAs of the zein nanofiber samples on the glass slide were measured using an OCA 25 contact angle meter (DataPhysics Instruments GmbH, Filderstadt, Germany). Briefly, 3 μL of distilled water was pipetted onto the samples, and the WCA was recorded at t = 5 s after the waterdrop had made contact with the nanofiber samples. Five tests were performed for each nanofiber sample.

### 2.6. Attenuated Total Reflectance Infrared Spectroscopy (ATR-FT-IR)

The FT-IR spectra of the zein nanofiber samples were collected using a Nicolet 6700 FT-IR spectrometer (Thermo Scientific, Waltham, MA, USA) with an ATR unit attached. All spectra in the wavenumber range of 4000–400 cm^−1^ were obtained by averaging 32 scans at 4 cm^−1^ resolutions.

### 2.7. Preparation and GTA Crosslinking of Zein Nanofibers Loaded with EU

Zein was dissolved in acetic acid (30%, *w*/*v*) at 25 ± 2 °C under continuous magnetic stirring. EU was added to the zein solution at a level of 20% (*w*/*w*) of zein content and allowed to mix for 1 h. After standing for 20 min, the mixture solution was electrospun using an electrospinning device (HZ-11, Huizhi Electrospinning (HZE), Qingdao, China) equipped with an applied voltage of 16 kV at a constant flow rate of 1 mL/h, a target roll speed of 1000 rpm, and a needle tip to collector distance of 10 cm. The electrospinning zein nanofibers were labeled EU/ZNF. Then, the zein nanofiber samples were crosslinked with GTA vapors for 3, 6, 9, and 12 h, which were labeled EU/ZNF_3h, EU/ZNF_6h, EU/ZNF_9h, and EU/ZNF_12h, respectively.

The EU contents in the zein nanofiber samples were determined according to the method of Aydin et al. (2022), with some modifications [48]. In brief, each nanofiber sample (15 mg) was placed in a dialysis bag containing 4 mL of ethanol aqueous solution (90%, *v*/*v*). The dialysis bag was then immersed in 60 mL of ethanol aqueous solution (90%, *v*/*v*) in a hermetic beaker and shaken at 25 °C and 80 rpm for 12 h. The concentration of EU in the release medium was then determined using a spectrophotometer at a wavelength of 280 nm. The encapsulation efficiency and loading capacity were calculated as follows:EE (%) = (Mass of actual EU loaded in nanofibers)/(Mass of nanofibers) × 100%(4)
LC (%) = (Mass of actual EU loaded in nanofibers)/(Mass of EU used in nanofibers) × 100% (5)

### 2.8. Immersion Study

To better understand the EU release behavior, the swelling characteristics of the zein nanofiber samples were determined according to the method of Martin et al. (2022), with slight modifications [49]. The zein nanofiber samples were cut into 3 cm × 3 cm (length × width) pieces. Subsequently, the samples were immersed in distilled water at room temperature for 0.5, 1, 3, 6, and 24 h, before photographing to measure the volumetric changes.

Non-crosslinked and crosslinked zein nanofibers loaded with EU were immersed in phosphate-buffered saline (PBS; pH = 7.2) at room temperature for 0.5, 1, 3, and 24 h, and then freeze-dried. Subsequently, their morphology was observed using a scanning electron microscope (Hitachi, Tokyo, Japan).

### 2.9. In Vitro Release Behavior of EU

The effect of crosslinking time on the release behavior of EU from zein nanofiber samples was investigated according to the method described by Chng et al. (2023) with some modifications [50]. The nanofiber samples (20 mg) were placed in a dialysis bag containing 4 mL of PBS (pH = 7.2). The dialysis bag was then immersed in 60 mL of PBS in a beaker and shaken at 25 °C and 80 rpm for 30 h. At specific time intervals (0.5, 1, 2, 3, 4, 5, 6, 8, 10, 12, 20, 24, and 30 h), 5 mL of the release medium was removed and replenished with an equal amount of fresh medium (PBS, pH = 7.2). The concentration of EU in the release medium was determined using a spectrophotometer at 280 nm.

### 2.10. EU Release Kinetics

As shown in Table 1, mathematical models, including zero-order, first-order, Higuchi, Ritger–Peppas, Peppas–Sahlin, and Kopcha, were used to fit the EU release curves to investigate the release mechanism of EU from the zein nanofiber samples.

### 2.11. Statistical Analysis

All determinations were performed at least in triplicate, and the results are presented as the mean ± standard deviation. Single-factor analysis of variance and Duncan’s multiple range test were used for statistical analysis, with a significance level of 0.05 (*p* < 0.05).

## 3. Results and Discussion

### 3.1. Morphology and Fiber Diameter Distribution

The morphology and fiber diameter distribution of the non-crosslinked and crosslinked zein nanofibers are shown in Figure 1 and Appendix A. According to Figure 1a, the average fiber diameter of GTA_0h was measured as 208.22 ± 46.93 nm; the average fiber diameters of GTA_3h, GTA_6h, GTA_9h, and GTA_12h were calculated as 225.52 ± 47.25 nm, 224.32 ± 52.93 nm, 226.96 ± 68.00 nm, and 267.69 ± 90.97 nm, respectively. The average fiber diameter and diameter distribution slightly increased with increasing exposure time to GTA vapor. The SEM images show that the fibers of GTA_0h were continuous and uniform. After crosslinking with GTA vapor, the fibers swelled but retained their fiber morphology, which is considered to be related to the absorption of water vapor during exposure to GTA vapors [35]. These results are consistent with those of previous studies, in which the fiber diameter of electrospun nanofibers increased after vapor-phase GTA crosslinking [35,52].

### 3.2. Mechanical Characterization

The mechanical properties of the non-crosslinked and crosslinked zein nanofibers are presented in Figure 2. The results indicated that GTA_0h showed minimal TS (11.08 ± 0.29 MPa). After crosslinking, the TS was observed to increase, with values of 15.28 ± 0.58 MPa, 17.59 ± 0.83 MPa, 23.60 ± 2.79 MPa, and 28.65 ± 0.43 MPa for GTA_3h, GTA_6h, GTA_9h, and GTA_12h, respectively. This increase might be due to the formation of chemical bonds among zein molecules inside and among the fibers, resulting from crosslinking during exposure to GTA vapors [35,53]. The longer crosslinking time would allow the generation of a denser network structure, improving the mechanical strength of the zein nanofibers. On the other hand, the YM of GTA-3h was 0.63 ± 0.06 GPa; after 12 h of vapor-phase GTA crosslinking, the YM value gradually increased to 0.88 ± 0.01 GPa, indicating that the zein nanofibers showed increased resistance to deformation with an increase in the crosslinking time. Notably, compared with the crosslinked zein nanofibers, GTA-0h had the most considerable YM value of 0.97 ± 0.05 GPa. These findings demonstrate that crosslinking decreases the rigidity of electrospun zein nanofibers. In addition, the EB increased continuously from 4.85 ± 0.86% of GTA-0h to 6.07 ± 0.12% of GTA-6h and then slightly decreased to 4.63 ± 0.36% of GTA-12h. One possible reason for this is that after 6 h of GTA crosslinking, the strong bond or three-dimensional structure formed between zein and GTA molecules would restrict the motion of the molecular chains [54,55]. Chen et al. (2022) fabricated electrospun feather keratin/gelatin nanofibers crosslinked by vapor-phase GTA and found that proper crosslinking could effectively enhance their mechanical strength [31]. Wang et al. (2016) also reported that the TS value of electrospun starch nanofibers was increased by ~10 times via crosslinking with vapor-phase GTA [52].

### 3.3. Water Contact Angle (WCA)

The WCA was used to investigate the surface hydrophilicity of the zein nanofibers before and after vapor-phase GTA crosslinking. The surface of nanofibers is considered hydrophobic when θ > 90°; on the contrary, it is considered a hydrophilic surface [56]. As shown in Figure 3, GTA_0h had a WCA of 120.88 ± 2.24°, indicating a hydrophobic surface. When the exposure time to GTA vapors was shorter than 9 h, the WCA values increased with exposure time. The WCA of GTA_6h was 126.38 ± 2.44°. This increased hydrophobicity can be attributed to the consumption of hydrophilic groups in zein molecules during crosslinking and the decrease in porosity of the crosslinked zein nanofibers, which restricted water penetration [31,55]. However, no significant difference in the WCA of the zein nanofibers was observed until the crosslinking time reached up to 12 h. After 12 h of vapor-phase GTA crosslinking, the WCA of the zein nanofibers decreased to 114.70 ± 2.87°, which may be related to their decrease in surface roughness and thickness. Notably, GTA_12h still possessed a hydrophobic surface (θ > 90°).

### 3.4. Stability

As a hydrophobic protein macromolecule, zein is insoluble in water. To examine the water resistance of the electrospun zein nanofiber samples, we used distilled water to immerse the as-spun and crosslinked zein nanofibers. Figure 4 and Figure 5 separately show the macroscopic images and the volume of non-crosslinked and crosslinked zein nanofibers after 24 h of immersion. GTA_0h was observed to shrink immediately when in contact with water, with a volume of 59.37% of the initial value. In comparison, the crosslinked zein nanofibers possessed better water resistance, with volumes of 64.07% (GTA_3h), 78.36% (GTA_6h), 84.39% (GTA_9h), and 69.58% (GTA_12h) upon exposure to water. When the immersion time was increased to 24 h, a decrease in the volumes of all samples was observed. The volumes of GTA_3h, GTA_6h, GTA_9h, and GTA_12h were 26.80%, 34.73%, 33.90%, and 35.17% of their original values, respectively, which is still higher than that of GTA_0h (28.34%). Furthermore, Appendix A shows the macroscopic images of non-crosslinked and crosslinked zein nanofibers upon immersion in acetic acid. GTA_0h was found to be fully dissolved in acetic acid. The crosslinked samples could better retain their integrity without dissolution. These results indicate that vapor-phase GTA crosslinking could improve the resistance of zein nanofibers to water and acetic acid. Yu et al. (2020) reported similar results, in that the moisture resistance of zein/PVA nanofibers was greatly enhanced after steam crosslinking with GTA [57]. Moreover, Selling et al. (2008) reported that in-site GTA-crosslinked zein nanofibers could dissolve in acetic acid [58]. The different results may be related to the difference in crosslinking methods and the amount of GTA used.

### 3.5. Fourier Transform Infrared Spectrometry (FT-IR)

The FT-IR spectra were recorded to explore the structural changes in the zein nanofibers before and after crosslinking with GTA vapors. As shown in Figure 6 and Appendix A, the spectra of GTA_0h presented a broad characteristic peak at 3295 cm^−1^, corresponding to the O–H and N–H stretching vibrations [59]. As the crosslinking time increased, that peak exhibited a redshift, and its peak intensity gradually decreased, suggesting the success of the crosslinking reaction. The free amino groups in the fibers reacted with the aldehyde groups in GTA molecules, leading to fewer free amino groups remaining to form O–H and N–H hydrogen bonds [60]. Chen et al. (2022) also reported similar results in electrospun feather keratin/gelatin nanofibers crosslinked by GTA vapors [31]. In addition, with increasing crosslinking time, the characteristic peaks representing the C=O stretching vibration, C–N stretching vibration, and N–H bending vibration (amide I band) were red-shifted from 1653 cm^−1^ of GTA_0h to 1648 cm^−1^ of GTA_12h [61,62]. The ATR-FT-IR spectra of all nanofiber samples showed the maximum amide II band at 1542 cm^−1^. After crosslinking, the characteristic maximum of the amide I/amide II bands shifted to lower frequencies from 1653 cm^−1^/1542 cm^−1^ of GTA_0h to 1648 cm^−1^/1542 cm^−1^ of GTA_12h. These results reflect the change in random coil structures, indicating the greater structural stability of crosslinked zein nanofibers than GTA_0h [63]. For the crosslinked zein nanofibers, a new characteristic peak appeared at ~1066 cm^−1^, corresponding to the symmetric stretching vibration of the C=O groups [64]. These findings confirm the crosslinking of zein nanofibers by GTA vapors. Overall, the crosslinked zein nanofibers showed secondary structures similar to those of as-spun zein nanofibers, suggesting that crosslinking post-treatment by GTA vapors on zein nanofibers had little effect on their secondary structure.

Figure 7 presents the possible molecular structure of zein after electrospinning and crosslinking. Acetic acid is a weak organic acid that is capable of breaking hydrophobic interactions, unfolding the β-sheet structure, and causing partial denaturation of the α-helix structure of proteins [65,66]. Zein was demonstrated to dissolve in acetic acid, with positive charges and a highly extended structure [66]. More free amino groups in the protonated zein molecules were exposed after electrospinning because of the rapid and large proportion of stretching. Moreover, fast solvent evaporation caused rapid solidification of the fibers, leading to free amino groups in zein being distributed on both sides of the molecular chains [52]. During exposure to GTA vapors, free amino groups were allowed to react with aldehyde groups in GTA molecules to form chemical bonds, which resulted in increased mechanical strength and water resistance of the crosslinked zein nanofibers [17].

### 3.6. In Vitro EU Release Profiles

As a model bioactive component, EU was incorporated into zein nanofibers. The encapsulation efficiencies of EU in EU/ZNF, EU/ZNF_3h, EU/ZNF_6h, EU/ZNF_9h, and EU/ZNF_12h were 95.61 ± 0.08%, 95.41 ± 0.12%, 95.73 ± 0.14%, 95.39 ± 0.16%, and 95.47 ± 0.11%, respectively. The corresponding EU loading capacities were separately 13.54 ± 0.05%, 13.43 ± 0.07%, 13.50 ± 0.09%, 13.63 ± 0.11%, and 13.39 ± 0.12%. This demonstrated that EU encapsulation in electrospun zein nanofibers was not affected by vapor-phase GTA crosslinking.

The release behavior of uncrosslinked and crosslinked zein nanofibers was then determined in PBS at 25 °C. According to Figure 8, all nanofiber samples exhibited a similar trend of EU release, comprising a burst release in the initial 2 h, followed by a sustained release up to 30 h. The initial release percentage for EU/ZNF was 26.15 ± 0.41%. The crosslinked zein nanofibers showed a decreased release percentage, with values of 23.31 ± 1.60% (EU/ZNF_3h), 18.15 ± 2.76% (EU/ZNF_6h), 16.66 ± 0.89% (EU/ZNF_9h), and 15.58 ± 2.23% (EU/ZNF_12h). At 30 h of release, approximately 81.38 ± 0.71% of release occurred from EU/ZNF. For EU/ZNF_3h, EU/ZNF_6h, EU/ZNF_9h, and EU/ZNF_12h, the cumulative release percentages were 74.80 ± 1.03%, 64.79 ± 2.58%, 59.89 ± 2.55%, and 54.74 ± 1.66% at 30 h, respectively. The initial burst release is possibly due to the rapid dissolution of EU close to the fiber surface, and the subsequent gradual release mostly depends on the diffusion rate of EU inside the fibers to the surface [33]. These results indicate that crosslinking post-treatment by GTA vapors could restrict molecular movement, ultimately extending the release time of EU from the zein nanofibers [67]. Furthermore, crosslinking reduced the number of 3D porous structures, which could restrict the permeation of water molecules, making it more difficult to release EU from crosslinked nanofibers [35]. Similarly, previous studies have reported that crosslinking treatment may prolong drug release from nanofibers [68,69]. Notably, the release rate of EU significantly decreased with the crosslinking time increasing to 6 h. Subsequently, a slower decline in the EU release rate was observed until 12 h of crosslinking. This decrease could be due to sufficient crosslinking after 6 h of the crosslinking reaction, resulting in a decline in the sustained release effect with the crosslinking time up to 12 h.

It has been demonstrated that fiber morphology can influence the drug-release behavior of electrospun nanofibers [70,71]. To better understand the release behavior of EU from crosslinked zein nanofibers by GTA vapors, the morphology of the nanofibers was observed after immersion in PBS (pH = 7.2) for 24 h. As shown in Figure 9, neat zein nanofibers exhibited poor stability in PBS, as EU/ZNF collapsed into a film after 0.5 h of immersion in PBS, likely due to the high hydrophobicity of the zein nanofibers. When the nanofibers were immersed in PBS, strong hydrophobic interactions between the fibers caused the formation of film-like substances [67]. The crosslinked zein nanofibers maintained their fiber shape and 3D porous structure in PBS after immersion for 24 h, although a certain extent of swelling of the fibers was observed. By increasing the crosslinking time up to 12 h, the stability of the nanofibers increased, which corresponds to the decrease in their volume shrinkage observed in Figure 5. This increased stability could be ascribed to the robust fiber matrix resulting from the formation of chemical bonds during crosslinking, resulting in their sustained-release performance. Similarly, Luo et al. (2018) fabricated gelation nanofibers crosslinked by GTA, which showed good stability in PBS [67]. In the study by Chen et al. (2022), improved water stability of feather keratin/gelatin nanofibers after vapor-phase GTA crosslinking was also observed [31].

To estimate the release mechanism of EU from non-crosslinked and crosslinked zein nanofibers, the corresponding release data of 60% dissolution were fitted to the zero-order, first-order, Higuchi, Ritger–Peppas, Peppas–Sahlin, and Kopcha models, and the results are presented in Table 2 [54]. The value of R^2^ in the first-order model (0.97–0.98) is obviously higher than that in the zero-order model (0.64–0.77), indicating that the EU release rate is matrix diffusion-controlled and dependent on EU concentration. The Peppas–Sahlin model was the best model to describe the release profile of EU from all zein nanofibers (R^2^ = 0.98–0.99). In the Peppas–Sahlin model, k_1_/k_2_ > 1 indicates that EU was released through a Fickian diffusion mechanism, k_1_/k_2_ < 1 indicates mainly an erosion mechanism, and k_1_/k_2_ = 1 indicates a combination of diffusion and erosion mechanisms. Based on the results shown in Table 2, the estimated K_1_/K_2_ ratios of EU/ZNF, EU/ZNF_3h, EU/ZNF_6h, EU/ZNF_9h, and EU/ZNF_12h were all higher than one. Moreover, the release exponent (n) estimated according to the Ritger–Peppas model for all samples was lower than 0.45, and the values of a/b estimated according to the Kopcha model for all samples were lower than one. Therefore, the release of EU from the as-spun and crosslinked zein nanofibers mainly followed the Fickian diffusion mechanism. Given that both EU and zein are insoluble in water, the diffusion of EU through the fibers is the main barrier to release. Surendranath et al. (2023) also indicated that the release of propranolol hydrochloride from thermally crosslinked zein/PVP nanofibers was mainly via Fickian diffusion [54].

## 4. Conclusions

In this study, crosslinking post-treatment with GTA vapors was used to improve the physicochemical properties and release performance of electrospun zein nanofibers. Compared with as-spun zein nanofibers, crosslinked samples retained their fiber structure, but the fiber diameter increased and the morphology became more compact. The surface hydrophobicity of the zein nanofibers did not change after crosslinking with GTA vapors for 12 h. The appropriate degree of crosslinking plays an essential role in improving the stability and mechanical properties of zein nanofibers. After 6 h of crosslinking with GTA vapors, the zein nanofibers exhibited a volume of 84.39% upon exposure to water, insolubility after 24 h of soaking in acetic acid, and a TS value of 23.60 ± 2.79 MPa. Exposure to GTA vapors resulted in the formation of new chemical bonds between the free amino groups in zein molecule chains and the aldehyde groups in GTA molecules, which increased the structural stability of the zein nanofibers. Moreover, vapor-phase GTA-crosslinked zein nanofibers showed a controlled release of EU in PBS. The model fitting results indicated that Fickian diffusion was the dominant release mechanism of EU, and the Peppas–Sahlin model is effective for describing the EU release behavior (R^2^ = 0.98–0.99). Accordingly, these crosslinked zein nanofibers have the potential to be used as active food packaging with sustained release. Further studies are underway to explore in vitro functional activities (antioxidant activity, antimicrobial activity), microbial penetration assay, cell cytotoxicity, the mechanism of EU–zein interaction, and the preservation effect on fruits.

## Figures and Tables

**Figure 1 foods-13-01583-f001:**
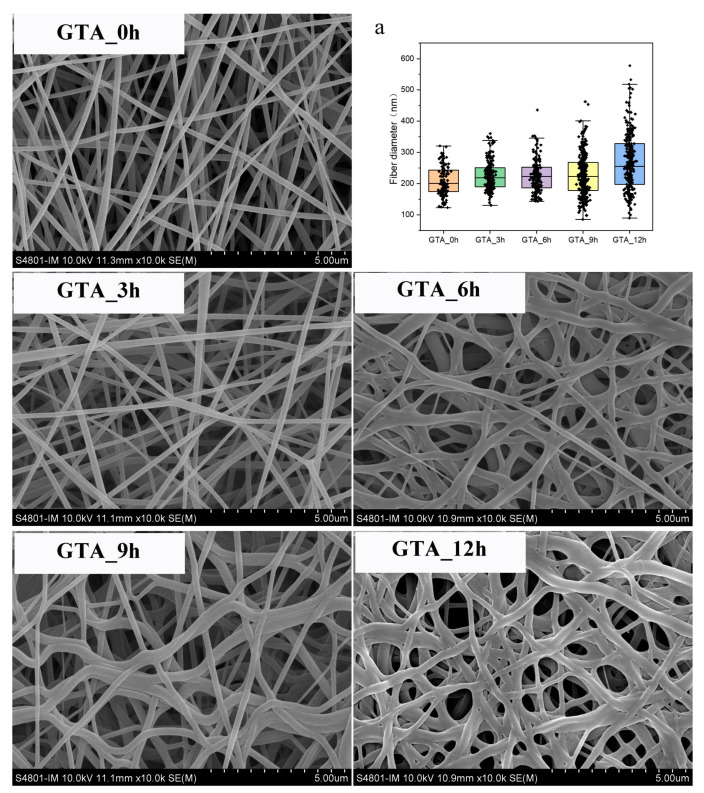
Scanning electron microscopy images and diameter distribution (**a**) of the as-spun and crosslinked zein nanofibers.

**Figure 2 foods-13-01583-f002:**
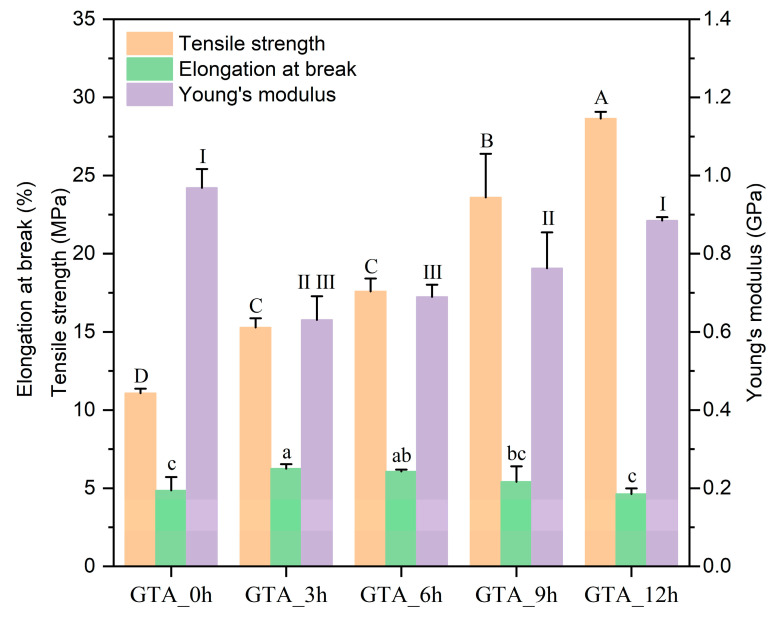
Mechanical properties of as-spun and crosslinked zein nanofibers. Various letters (A–D, a–c) and Roman letters on the top of the columns suggest the significant difference (*p* < 0.05).

**Figure 3 foods-13-01583-f003:**
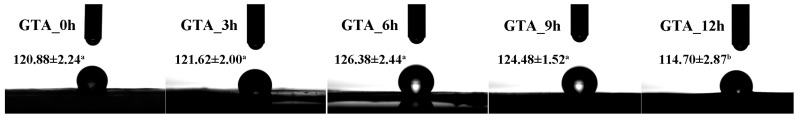
Water contact angle of the as-spun and crosslinked zein nanofibers. Various lowercase letters suggest the significant difference (*p* < 0.05).

**Figure 4 foods-13-01583-f004:**
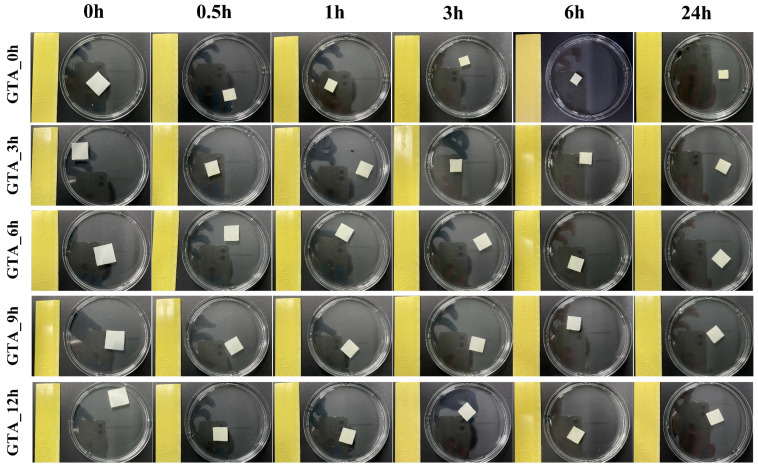
Macroscopic images of the as-spun and crosslinked zein nanofibers after immersion in water for 24 h.

**Figure 5 foods-13-01583-f005:**
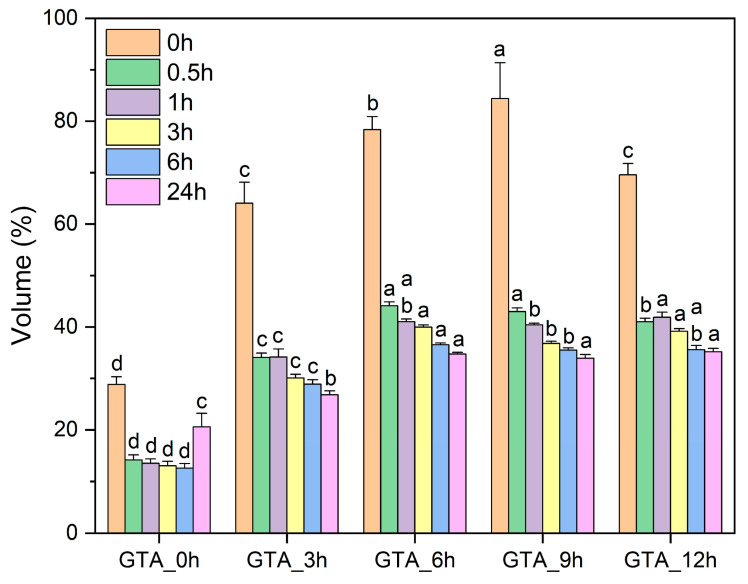
Volume changes in the as-spun and crosslinked zein nanofibers after immersion in water for 24 h. Various lowercase letters on the top of the columns suggest the significant difference (*p* < 0.05).

**Figure 6 foods-13-01583-f006:**
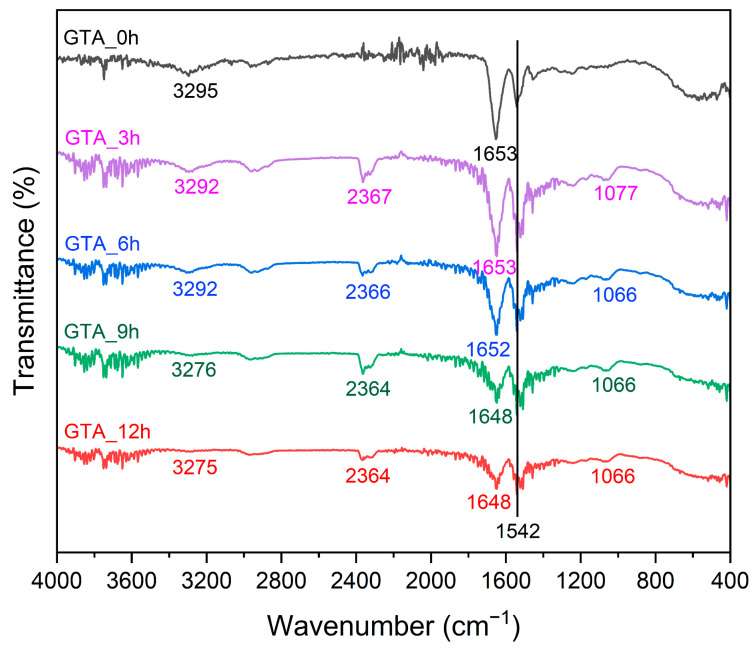
FT-IR spectra of the as-spun and crosslinked zein nanofibers.

**Figure 7 foods-13-01583-f007:**
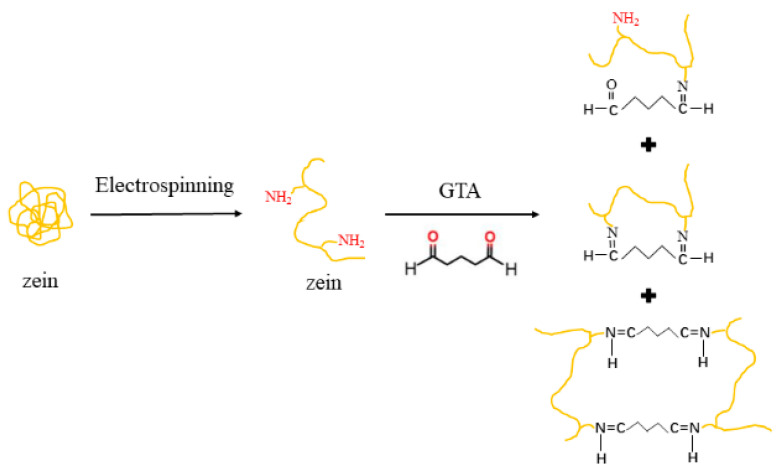
Effect of electrospinning and GTA crosslinking on the molecular structure.

**Figure 8 foods-13-01583-f008:**
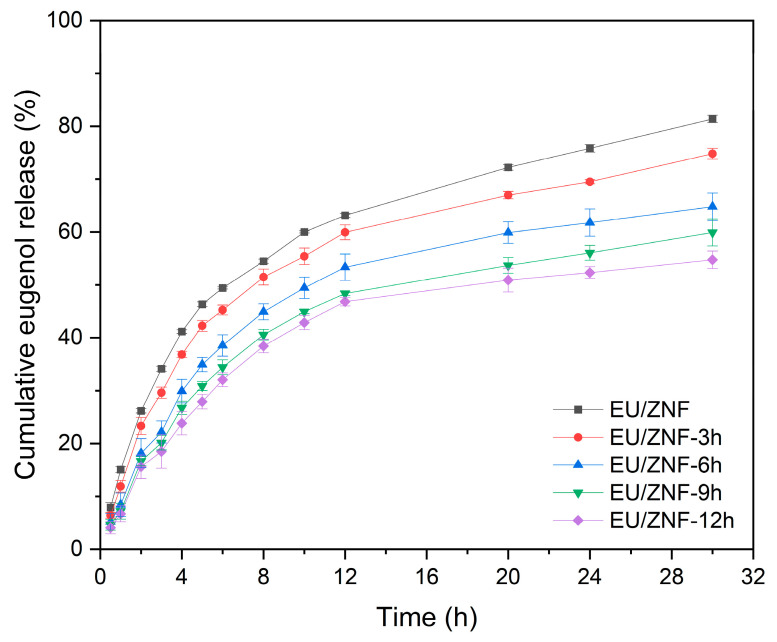
In vitro release profiles of eugenol from non-crosslinked and crosslinked zein nanofibers in PBS.

**Figure 9 foods-13-01583-f009:**
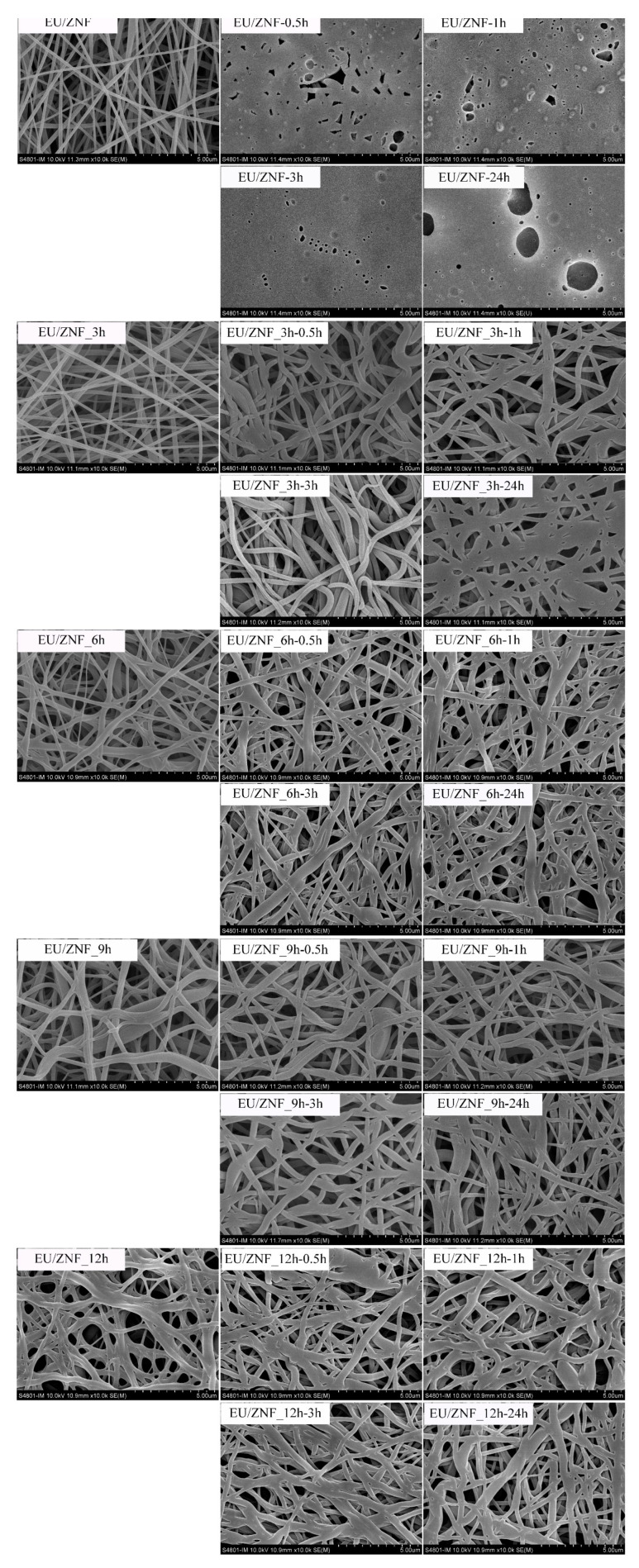
Scanning electron microscopy images of eugenol-loaded zein nanofibers crosslinked with vapor GTA after immersion in PBS for 24 h.

**Table 1 foods-13-01583-t001:** Mathematical models for the analysis of the eugenol release kinetics from zein nanofibers [51].

Model	Equation	Parameter
Zero-order	y = a + b × x	-
First-order	y = a × (1 − exp(−b × x))	-
Higuchi	y = k × x^0.5^	k: release constant; x: time
Ritger–Peppas	y = k × x^n^	k: release constant; x: time; n: release exponent (n ≤ 0.45: Fickian diffusion; 0.45 < n < 0.89: non-Fickian diffusion; n ≥ 0.89: erosion).
Peppas–Sahlin	y = k_1_ × x^m^ + k_2_ × x^(2×m)^	k_1_: diffusion constant; k_2_: erosion constant; x: time; m: Fickian diffusion exponent; k_1_/k_2_ > 1: mainly Fickian diffusion; k_1_/k_2_ < 1: mainly erosion; k_1_/k_2_ = 1: Fickian diffusion and erosion.
Kopcha	y = a × x^0.5^ + b × x	a/b > 1: mainly Fickian diffusion; a/b < 1: mainly erosion; a/b = 1: Fickian diffusion and erosion.

**Table 2 foods-13-01583-t002:** Fitting parameters of the release model for non-crosslinked and crosslinked zein nanofibers loaded with eugenol.

Model	Parameter	EU/ZNF	EU/ZNF_3h	EU/ZNF_6h	EU/ZNF_9h	EU/ZNF_12h
Zero-order	a	37.56 ± 5.29	30.38 ± 4.79	30.34 ± 4.48	31.30 ± 4.27	29.72 ± 3.78
	b	2.19 ± 0.40	2.23 ± 0.36	1.84 ± 0.34	1.43 ± 0.32	1.30 ± 0.29
	R^2^	0.73	0.77	0.73	0.64	0.65
First-order	a	83.78 ± 2.16	79.69 ± 1.81	69.75 ± 1.46	60.08 ± 1.14	55.50 ± 1.29
	b	0.24 ± 0.02	0.19 ± 0.01	0.23 ± 0.01	0.31 ± 0.02	0.33 ± 0.03
	R^2^	0.97	0.98	0.98	0.98	0.97
Higuchi	K	20.03 ± 0.99	18.09 ± 0.71	16.46 ± 0.80	15.05 ± 0.96	14.04 ± 0.90
	R^2^	0.79	0.89	0.81	0.59	0.55
Ritger–Peppas	k	30.93 ± 2.42	24.81 ± 2.04	24.90 ± 2.06	26.02 ± 2.30	24.81 ± 1.98
	n	0.34 ± 0.03	0.38 ± 0.03	0.34 ± 0.03	0.29 ± 0.03	0.28 ± 0.03
	R^2^	0.94	0.95	0.94	0.9	0.91
Peppas–Sahlin	k_1_	27.86 ± 1.72	21.32 ± 1.30	21.63 ± 1.11	23.82 ± 1.75	23.31 ± 1.56
	k_2_	−2.21 ± 0.28	−1.38 ± 0.17	−1.61 ± 0.17	−2.23 ± 0.35	−2.30 ± 0.32
	m	0.56 ± 0.03	0.62 ± 0.03	0.59 ± 0.03	0.54 ± 0.04	0.51 ± 0.04
	R^2^	0.99	0.99	0.99	0.98	0.98
Kopcha	a	30.63 ± 1.02	25.23 ± 1.09	24.94 ± 0.87	25.37 ± 0.87	23.86 ± 0.72
	b	−2.61 ± 0.24	−1.76 ± 0.26	−2.08 ± 0.20	−2.54 ± 0.20	−2.41 ± 0.17
	R^2^	0.98	0.98	0.98	0.97	0.98

## Data Availability

The original contributions presented in the study are included in the article/Appendix A, further inquiries can be directed to the corresponding author.

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
