# Peer review of "Improvement in the Sustained-Release Performance of Electrospun Zein Nanofibers via Crosslinking Using Glutaraldehyde Vapors"

_foods, 2024, doi:10.3390/foods13101583_

Round 1
Reviewer 1 Report
Comments and Suggestions for Authors
The submitted manuscript presents the fabrication of zein nanofibers by electrospinning and their crosslinking with glutaraldehyde (GA) vapors. The obtained fibrous materials were characterized with scanning electron microscopy (SEM), mechanical properties, water resistance, and Fourier transform infrared (FTIR) spectroscopy. The authors stated that increasing the crosslinking time resulted in improvement of the mechanical properties and their water resistance.
Specific points requiring attention that should be addressed by the authors are detailed below.
1. The Introduction is not comprehensive. It does not contain summarized information concerning the electrospinning of zein and characterization of zein nanofibers. There are a lot of such investigations.
2. The selected crosslinker – GA and even its vapors have irritant effect to the skin, eyes and respiratory system.
3. In the cited reference [11] there are no results showing the lack of GA toxicity.
4. The shown fibers presented by SEM micrographs in Figure 1 should be captured at higher magnification (x5000). From the used magnification the fiber morphology is hardly visible and could not be used to measure the fiber diameters and distribution using Image J.
5. The presented accuracy in fibers diameter distribution is doubtful. The values should be rounded.
6. The captured water droplets are so miniature. Show more representative and easily visible images.
7. Figure 4 is much overburdened. It is easily visible that the shown samples in first line are not precisely and equally cut.
8. The magnification of fibrous samples shown in Figure9 is not the proper one.
Comments on the Quality of English LanguageModerate editing of English language required
Reviewer 2 Report
Comments and Suggestions for Authors
Overall the manuscript is ok.
1. The author needs to improve the introduction part with more information on Eugenol and its medicinal values.
2. Besides the author also needs to describe why this current study is importance and what is the significance of drug release parameters.
3. The author can make a table of different functional groups and structural changes present in zein nanofibers and why these are important.
4. Why this study is important from a food point of view should be clarified.
Comments on the Quality of English Language
Minor editing of English language required.
Round 2
Reviewer 1 Report
Comments and Suggestions for Authors
The authors addressed the questions and suggestions. The article could be accepted in the revised form.
Reviewer 2 Report
Comments and Suggestions for Authors
The author have made necessary changes.